# Therapeutic Potential of VEGF-B in Coronary Heart Disease and Heart Failure: Dream or Vision?

**DOI:** 10.3390/cells11244134

**Published:** 2022-12-19

**Authors:** Rahul Mallick, Seppo Ylä-Herttuala

**Affiliations:** 1A.I. Virtanen Institute for Molecular Sciences, University of Eastern Finland, 70211 Kuopio, Finland; 2Heart Center and Gene Therapy Unit, Kuopio University Hospital, 70029 Kuopio, Finland

**Keywords:** angiogenesis, CHD, gene therapy, VEGF-A, VEGF-B, VEGF-R1, NRP-1, heart failure, fatty acid, glucose, FATP

## Abstract

Coronary heart disease (CHD) is the leading cause of death around the world. Based on the roles of vascular endothelial growth factor (VEGF) family members to regulate blood and lymphatic vessels and metabolic functions, several therapeutic approaches have been attempted during the last decade. However proangiogenic therapies based on classical VEGF-A have been disappointing. Therefore, it has become important to focus on other VEGFs such as VEGF-B, which is a novel member of the VEGF family. Recent studies have shown the very promising potential of the VEGF-B to treat CHD and heart failure. The aim of this review article is to present the role of VEGF-B in endothelial biology and as a potential therapeutic agent for CHD and heart failure. In addition, key differences between the VEGF-A and VEGF-B effects on endothelial functions are demonstrated.

## 1. Introduction

Vascular endothelium regulates the flow of fluids and molecules into and out of tissues. The endothelium represents the luminal cellular monolayer of the vessels where endothelial cells are tightly connected, forcing most nutrients to be transported through the plasma membrane by specific transport systems. Endothelium also regulates vascular contractility and permeability [1]. Disruption of the endothelial homeostasis results in pathological conditions, one of which is coronary heart disease (CHD). The mismatch between myocardial oxygen supply and demand due to narrowing or blockage of coronary arteries causes coronary heart disease (CHD) (also known as “coronary artery disease” or “ischemic heart disease”) [2,3,4,5]. This notorious disease is the leading cause of death worldwide [2,3,5,6]. The lifetime risk of CHD at age 50 is higher for men [7]. CHD frequently leads to heart failure [8]. Pharmacotherapy and/or revascularization (percutaneous coronary intervention, coronary artery bypass grafting) therapy are the currently available treatment options [9]. However, many patients cannot be treated with these conventional methods due to insufficient subendocardial blood flow or no-flow condition [10]. Henceforth, novel therapeutic strategies are in high demand. Therapeutic angiogenesis could be a potential treatment option for CHD patients by stimulating blood vessel growth, improving tissue perfusion along with supporting tissue regeneration and recovery capacity [11,12].

Atherosclerosis is the most common cause of CHD. Several factors such as lipid metabolic disorders, chronic inflammation, endothelial cell dysfunction, and immune dysfunction promote atherosclerosis, and lead to CHD and finally to heart failure [12]. Clinically, dyslipidemia is present in around 70% of CHD patients and lipid lowering therapy significantly reduces the risk of CHD [13,14]. In addition, leukocyte recruitment and chronic inflammation lead to endothelial cell activation, monocyte influx, further chemotaxis of inflammatory cells, foam cell formation, and macrophage apoptosis in early steps of atherosclerosis [15]. Thus, improving lipid metabolism and alleviating inflammatory responses are key approaches for the current therapeutic strategies for CHD and heart failure.

Many angiogenic growth factors and transcription factors have been identified that possess useful therapeutic properties in cell proliferation, anti-apoptotic ability, and energy metabolism [11]. In this context, members of the vascular endothelial growth factor (VEGF) family are the most studied factors to induce angiogenesis, lymphangiogenesis as well as to regulate lipid metabolism, inflammation, and oxidative stress [12]. Figure 1 shows VEGFs and their receptors. However, several randomized clinical trials targeting VEGF-A to treat CHD and heart failure have faced setbacks and failed to meet expectations [11]. Therefore, new therapeutic strategies for CHD and heart failure patients by using new molecules, improved vectors as well as delivery methods are needed. As very few, if any, reviews about the newest member of the VEGF family, VEGF-B, on CHD and heart failure have been published recently, we summarized the current biology, functions, and therapeutic potential of VEGF-B for the treatment of CHD and heart failure.

## 2. Vascular Endothelial Growth Factor B

Following the discovery of the homolog of VEGF-A on chromosome 11q13.1 in 1995, VEGF-B was initially considered as an angiogenic growth factor. VEGF-B is highly expressed in metabolically active tissues including heart [16,17,18,19]. Other growth factors, hypoxia or hormones, do not regulate VEGF-B expression although recent findings have demonstrated the upregulation of VEGF-B expression by vagal neurotransmitter acetylcholine and resveratrol [19,20,21]. In addition, regulation of the methylation status of the VEGF-B promoter by nutriepigenetic modulation has been identified [22]. Genetic knockout of VEGF-B did not hamper normal health in mice [17]. Two alternatively spliced isoforms of VEGF-B have been identified: heparan sulfate proteoglycan binding domain containing VEGF-B167 (predominant isoform; over 80%) and freely soluble hydrophobic C-terminal containing VEGF-B186 [23,24,25] (Figure 2). Despite having a significant variation in the ratio of VEGF-B167/VEGF-B186, human tumor cell lines predominantly express VEGF-B186 [26].

The functions of VEGF-B are mediated by its binding to a tyrosine kinase receptor, named as vascular endothelial growth factor receptor 1 (VEGF-R1) and common co-receptors neuropilin 1 (NRP-1) or NRP-2 [27,28,29]. It needs to be pointed out that VEGF-B186 binds to VEGF-R1 before and after proteolytic cleavage but requires proteolytic cleavage to bind to NRP-1 or NRP-2 [28,30]. VEGF-A does not seem to significantly activate VEGF-R1 downstream signaling compared with VEGF-R2 mediated downstream signaling, while VEGF-B binding to VEGF-R1 leads to the activation of AKT, PI3K, ERK, and MAPK [31].

## 3. Cardioprotective Role of VEGF-B

The heart expresses a high level of VEGF-B [32,33]. Interestingly, cardiac VEGF-B expression changes from the right to left ventricular wall during embryonic and early post-natal periods [34,35]. In the heart, the highest expression of VEGF-B is found in cardiomyocytes and the lowest expression is in endothelial cells [36,37,38]. In contrast, VEGF-R1 is predominantly expressed in cardiac endothelial cells [36,37,38]. VEGF-B contributes to the cardiac remodeling process following myocardial infarction, and the VEGF-B level declines in heart failure and diabetic heart [39,40,41]. Thus, VEGF-B has a cardioprotective role in regulating myocardial contractility, metabolism, and protecting cardiomyocytes from ischemic and apoptotic damage via a physiological hypertrophy (Table 1) [42,43,44]. Cardioprotective mechanisms of VEGF-B are discussed below in relation to the treatment of CHD and heart failure along with the concerning challenges.

### 3.1. Unique Neovascularization in the Heart

VEGF-B promotes neovascularization in metabolically active tissues including heart. In connection with coronary vessels, VEGF-B promotes angiogenesis in the subendocardial region and rescues the myocardium after myocardial infarction [52]. Our studies have shown that adenoviral vector-based VEGF-B186 gene transfer induced significant angiogenesis in the myocardium and improved myocardial perfusion without increasing a coronary steal effect [28,43,45,55]. Thus, VEGF-B186 seems highly beneficial for the treatment of CHD [43,45]. However, it has remained unclear if both unprocessed and proteolytically processed forms or specific forms of VEGF-B186 contribute to the neovascularization process. To resolve these questions, we have designed several VEGF-B186 isoforms, based on its structure, proteolytic processing, and known receptor binding properties [45]. The studies have found that only the unprocessed full-length form of VEGF-B186 induces angiogenesis in the heart and NRP binding is dispensable for the VEGF-B induced microvascular growth in the heart muscle [28,45]. Interestingly, despite having VEGF-R1 binding ability, the N-terminal forms of VEGF-B186 (e.g., VEGF-B127 and VEGF-B109) did not show any angiogenic activity in the heart [45]. Recent results of Korpela et al. and Robciuc et al. confirmed a similar outcome of neovascularization in mice with endothelial cell-specific deletion of VEGF-R1 [56]. Actually, these results question the significance of VEGF-R1 in the VEGF-B induced angiogenic process. In addition, the current findings on the accumulation of the endothelial progenitor cells and their contribution to the neovascularization process in the heart following adenoviral gene transfer of VEGF-B186 and unsplicable VEGF-B186 (defined as “VEGF-B186R127S” due to a mutation in arginine to serine at 127 position of the amino acid sequence) as well as the diminished neovascularization effect of VEGF-B186 due to the binding to a soluble VEGF-R1 raises the possibility that there is a yet unidentified cell surface receptor for the C-terminal end of the full-length form of VEGF-B186, which could induce the neovascularization process [28,43]. Further studies are required to explore the exact mechanism of VEGF-B induced angiogenesis in the heart.

### 3.2. Anti-Apoptosis

VEGF-B has a strong anti-apoptotic effect on various cells including cardiomyocytes by activating VEGF-R1. Recombinant VEGF-B167 has shown to downregulate the expression of apoptotic genes (*BMF*, *BAD*, *BID*, *BAX, CASP9*, *DCN*, *TP53INP1*, *TNF*) more effectively than other VEGF-R1 ligands such as VEGF-A or placental growth factor (PlGF) [43,57]. Hyperglycemia releases VEGF-B167 from cell surface heparan sulfate proteoglycans, which is secondarily influenced by glucose-induced heparinase secretion from the endothelial cells [40]. Lal and colleagues subsequently showed that VEGF-B167 binding to VEGF-R1 attenuates apoptosis (H_2_O_2_ induced poly-(ADP-ribose) polymerase and caspase 3 activity) by activating ERK/glycogen synthase kinase-3β in cardiomyocytes and endothelial cells [40]. Collectively, glucose sensing ability by the endothelial cells also protects the heart via autocrine and paracrine actions of VEGF-B. Despite elevated VEGF-B levels in circulation and VEGF-R1 expression in the heart, diminished heparinase as well as VEGF-B levels in the diabetic heart may aggravate the cardiomyocyte dysfunction and/or susceptibility to death [40,58]. Therefore, targeting VEGF-B186 mediated signaling over VEGF-B167 could be a potential therapeutic strategy to treat CHD in the presence or absence of diabetes. Although the higher level of circulating VEGF-B raises the possibility of VEGF-B resistance in the diabetic heart, the role of VEGF-B in the progression of diabetic cardiomyopathy should be explored. Additionally, the potential therapeutic efficacy of VEGF-B186 in the diabetic heart should be studied further.

### 3.3. Antioxidant Activity

Oxidative stress increases the risk of CHD by damaging vascular endothelial cells. VEGF-B induces many antioxidant genes (*Gpx1*, Gpx2, *Sod1*, *Prdx1*, *Prdx5*, *Prdx6-rs1*, *Txnrd3*, *Sod2*, *Gpx5*, *Zmynd17*, etc.) and downregulates genes responsible for oxidative stress (*Ptgs1*, *Nox4*, *Nef2*, *Tpo*, *Ppp1r15b* etc.) [59]. VEGF-B also protects myocardium via the AMPK/eNOS/NO signaling pathway following myocardial infarction [20].

### 3.4. Distinct Metabolic Reprogramming

VEGF-B is co-expressed with a cluster of nuclear encoded mitochondrial genes involved in the oxidative phosphorylation mechanism [32]. Binding of VEGF-B to VEGF-R1 and NRP-1 promotes the distribution of fatty acid transport proteins (FATP3 and FATP4) on the endothelial cells to facilitate lipid transport where lipoprotein lipase and low-density lipoprotein receptor are merely anchored [25,32,43,53,60]. VEGF-B reduces endothelial cholesterol content by interfering with low-density lipoprotein receptor recycling [60]. It is noteworthy to mention that freely soluble VEGF-B186 is more effective than heparan sulfate proteoglycan bound VEGF-B167 in inducing FATP expression [25]. Trans-endothelial transport of circulating fatty acids for subsequent utilization by cardiomyocytes is promoted by VEGF-B [25,61]. However, VEGF-B signaling rapidly diminishes endothelial cholesterol uptake [60]. Excess myocardial fatty acid uptake and utilization leads to inhibition of pyruvate dehydrogenase due to a rise in mitochondrial acetyl-CoA levels and attenuates glucose oxidation and subsequent glycolysis (termed as “Randle cycle”) in the heart muscle by several pathways including insulin signaling [43,62,63]. Interestingly diminished endothelial glucose transcytosis through insulin insensitive glucose transporter 1 (predominantly expressed in the heart) and cardiac glucose utilization are co-ordinated with reduced membrane cholesterol level by VEGF-B signaling [60]. Despite reduced glucose uptake, how the glycogen accumulation is upregulated by VEGF-B is still a mystery [60]. The metabolic effects of VEGF-B seem to be mediated through the direct activation of VEGF-R1 and NRP-1 [60].

Metabolically active tissues including the heart muscle depend on fatty acids as the predominant source of energy [64]. In diabetes, the heart cannot effectively utilize glucose as an energy source and relies mostly on fatty acids for energy production [65]. Energy production through fatty acid oxidation requires more oxygen than glucose utilization and in diabetes, VEGF-A mediated angiogenesis is obscured due to the reduced oxygen supply [66,67]. Concerning insulin resistance, imbalance of fatty acid uptake and oxidation due to the lack of oxygen results in the accumulation of ceramides and diacylglycerols, which amends the effects of insulin mediated glucose uptake related effects on gene expression and signaling in the heart [1,62,68,69]. As VEGF-B is able to induce ceramide accumulation and mitochondrial dysfunction in the heart due to the imbalance of fatty acid uptake and oxidation [42], the therapeutic use of VEGF-B to treat heart failure in diabetes or dyslipidemia could also have disadvantages. However, it is noteworthy to mention that the fatty acid oxidation-related effects of VEGF-B on glucose metabolism were all reported from transgenic and knockout mice on a high-fat diet and were not always repeatable [18,32,42,70]. Thus, the possibility of a high-fat diet induced insulin resistance cannot be excluded [71,72]. Corroborating these findings, Kivelä et al. demonstrated the shift of fatty acid oxidation toward glucose metabolism by VEGF-B in the heart of transgenic rats [39]. Similarly, viral-vector mediated VEGF-B186 gene transfer into mouse adipose tissue supports beneficial outcomes such as insulin sensitivity, glucose tolerance, and improved metabolic health [56]. In addition, Tirronen et al. found reduced lipid deposition in the heart, but predisposition to heart failure by VEGF-B in transgenic mice [73]. Combining all the information, it can be suggested that due to the transient duration of the adenoviral vector-based VEGF-B186 gene therapy, this treatment might still be beneficial in heart failure patients with dyslipidemia.

### 3.5. Cardiac Hypertrophy

VEGF-B mediated cardiac hypertrophy is thought to be dependent on VEGF-R1 expression in cardiomyocytes [46,50,51,74]. Unlike PlGF, VEGF-B mediated cardiac hypertrophy is independent of nitric oxide mediated mechanism as well as NRP-1 mediated signaling [39,51]. It is noteworthy to mention that the expression of cytoplasmic bone marrow kinase in chromosome X in arterial endothelium is crucial for VEGF-B induced cardiac hypertrophy [51]. However, the possibility of the involvement of paracrine signaling between cardiomyocytes and endothelial cells cannot be excluded [43,51,75,76]. Involvement of cell types other than cardiomyocytes in the VEGF-B mediated cardiac hypertrophy has been suggested as α-smooth muscle actin-positive cells and endothelial cells show clear proliferative activity in the VEGF-B transgene treated heart [28,43].

### 3.6. Cardiac Contractility

It has been reported that VEGF-B can maintain cardiac contractility [39]. VEGF-B has been found to preserve cardiac function and to induce physiological hypertrophy. VEGF-B167 induces *α-MHC*, *SERCA2a*, *RYR*, *ANF*, *BNP*, and *PGC1α* genes and represses the expression of *β-MHC* and *α-actin 1* genes, which attenuate the loss of myocardial mass and maintain cardiac contractility [50]. Additionally, VEGF-B overexpression was found to have effects on cardiac electrophysiology (e.g., downregulation of sodium current (I_Na_), transient outward current (I_to_), and total K^+^ current (I_peak_)) as well as hinder Ca^2+^ signaling in the mouse heart [77]. VEGF-B186 transgene delivery has been shown to improve cardiac ejection fraction following myocardial infarction [52]. Recently, VEGF-B overexpressed mice (under dobutamine stress) as well as VEGF-B186 and soluble VEGF-R1 transgene co-delivery in pigs were shown to induce cardiac arrhythmias, suggesting the role of NRP-1 in the growth of sympathetic nerves and subsequent cardiac arrhythmias [54]. However, recent findings in mice and porcine studies using mutant unsplicable VEGF-B186 did not support the role of NRP-1 in provoking cardiac arrhythmias [28,45,52]. Instead, Korpela et al. reported that the C-terminal fragment of the proteolytically spliced VEGF-B186 induced cardiac arrhythmias [45]. As VEGF-R1 is known to regulate cardiac performance [78], it is possible that due to the inability to bind to the VEGF-R1, the C-terminal end of the VEGF-B186 provokes arrhythmias via binding to an unidentified receptor in the heart.

## 4. Differences between VEGF-A and VEGF-B Mediated Effects

Vascular endothelial cell proliferation and neovascularization processes are stringently dependent on cellular metabolism such as glycolysis and fatty acid oxidation. Agitation of the endothelial cell metabolism has been found to be linked with health and disease conditions (e.g., diabetes and tumorigenesis) [1]. Angiogenesis begins with endothelial cell activation by pro-angiogenic proteins such as VEGF-A and VEGF-B. VEGF-A activated endothelial cells extend toward angiogenic stimuli via forming filopodia (thin membrane protrusions) and tip cells (Figure 3) [79]. Migratory tip cells are followed by proliferating stalk cells to elongate the sprouting process [80]. Tip cells try to communicate with adjacent tip cells to form neovessel networks to meet oxygen and nutritional demands. VEGF-A induced activation of endothelial cells also increases vascular permeability [81]. With VEGF-B stimulation, no evidence of the formation of tip cells has been found. Rather, VEGF-B186 mediated activated endothelial cells recruit endothelial progenitor cells to form neovessels, which are then surrounded by pericytes to prevent vascular leakage [28,45,82].

VEGF-A mediated glucose uptake and utilization is crucial for the tip cell migration and navigation, whereas the proliferating stalk cell relies on fatty acid oxidation to sustain vessel growth [83,84,85,86,87,88,89]. The glycolytic signaling regulates VEGF-R2 and NOTCH-1 functions to balance angiogenic process [90,91]. On the other hand, VEGF-B186 reduces glucose uptake and utilization by activated endothelial cells [60], which may explain the limited tip cell activity. However, when endothelial progenitor cells are recruited, VEGF-B186 supports the stabilization of neovessels via fatty acid uptake and utilization [32,92]. It is noteworthy to mention that VEGF-B167 has been shown to have similar metabolic reprogramming effects as VEGF-B186 [32,60].

In addition, VEGF-B mediated angiogenesis and local inflammation are less prominent than those induced by VEGF-A [43,51].

## 5. Summary

In addition to its angiogenic role, the regulation of myocardial contractility and metabolism as well as the cardioprotective role of VEGF-B have made it distinct from the other pro-angiogenic factors of the VEGF family (Figure 4) [11]. Cumulatively, the above data suggest that VEGF-B is a strong candidate for the treatment of CHD and heart failure.

## Figures and Tables

**Figure 1 cells-11-04134-f001:**
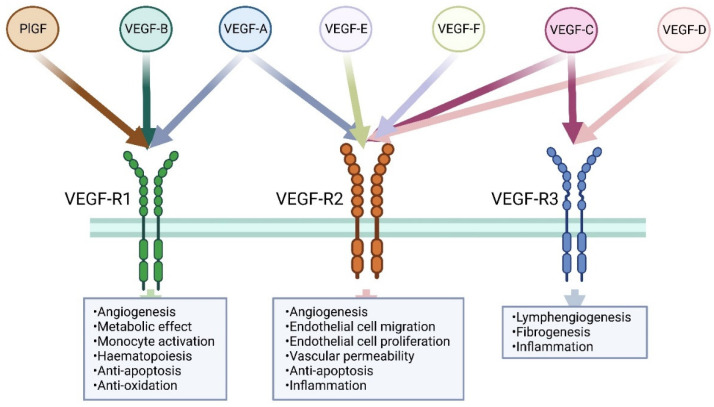
Members of the vascular endothelial growth factor family. Three VEGF receptors were identified. VEGF-R1 and VEGFR2 are predominantly expressed on vascular endothelial cells, while VEGFR3 is mainly expressed in lymphatic endothelial cells. VEGF-A, VEGF-B, and PlGF are the ligands for VEGF-R1. VEGF-A, VEGF-C, VEGF-D, VEGF-E (viral source), and VEGF-F (snake venom source) are the ligands for VEGFR2, while only VEGF-C and VEGF-D bind to VEGFR3.

**Figure 2 cells-11-04134-f002:**
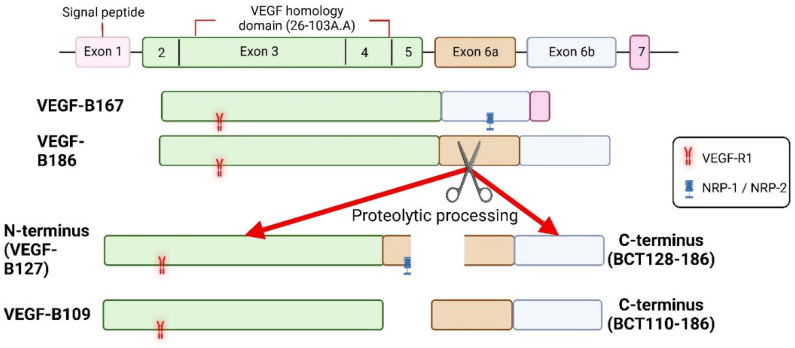
Isoforms of VEGF-B and known receptor binding sites. Both isoforms of VEGF-B bind to VEGF-R1 and NRPs, but VEGF-B186 binds to NRPs only after proteolytic processing. Receptor binding profile of VEGF-B186 C-terminus is not known.

**Figure 3 cells-11-04134-f003:**
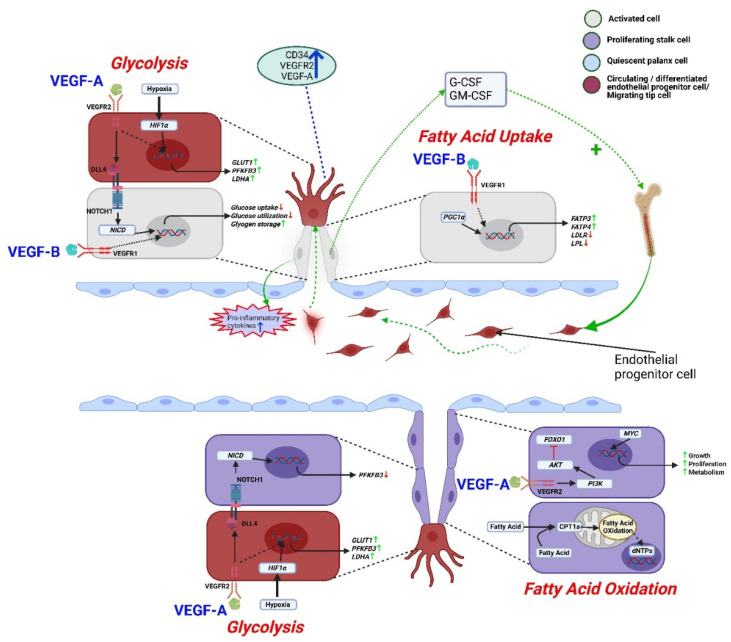
Regulation of endothelial metabolism by VEGF-B and VEGF-A. VEGF-B activates endothelial cells and secretes pro-inflammatory cytokines including G-CSF, GM-CSF to recruit endothelial progenitor cells. VEGF-A secreting differentiated endothelial progenitor cells overexpress genes like *LDHA*, *PFKFB3*, and *GLUT1* through the VEGF-A-VEGFR2 mediated glycolytic pathway. A similar signaling pathway was observed in tip cells to support migration. Hypoxia can also increase glycolysis by stimulating HIF1α. The glycolysis signaling supports stalk cell proliferation by downregulating DLL4-NOTCH1 signaling-dependent PFKFB3 gene expression. On the other hand, VEGF-B-VEGF-R1 mediated signaling uniquely decreases glucose uptake and utilization, but increases fatty acid uptake as well as glycogen storage to sustain activated endothelial cell survival. Furthermore, VEGFA-VEGFR2 mediated PI3K/AKT pathway inhibits growth-inhibiting transcription factor FOXO1 to support the proliferation of stalk cells. Growth-enhancing transcription factor MYC promotes growth, anabolic metabolism, and proliferation of stalk cells. Along with glucose, proliferating stalk cells utilize fatty acids, contributing to nucleotide synthesis for cell proliferation. Here, AKT: protein kinase B, CPT1a: carnitine palmitoyltransferase 1a, dNTP: deoxynucleoside triphosphate, DLL4: delta-like 4, FOXO1: forkhead box protein O1, G-CSF: granulocyte colony stimulating factor, GM-CSF: granulocyte-macrophage colony-stimulating factor, GLUT1: Glucose transporter 1, HIF1α: hypoxia-inducible factor 1-alpha, LDLR: low density lipoprotein receptor, LPL: lipoprotein lipase, LDHA: lactate dehydrogenase A, NOTCH1: Notch Receptor 1, NICD: notch intracellular domain, PFKFB3: 6-phosphofructo-2-kinase/fructose-2,6-biphosphatase 3, PI3K: phosphoinositide 3-kinase.

**Figure 4 cells-11-04134-f004:**
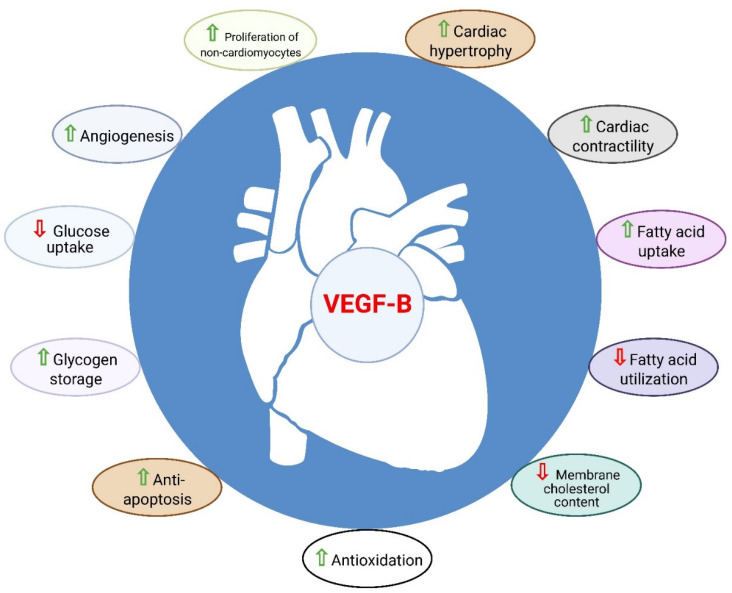
Role of VEGF-B in the heart. VEGF-B is known to have cardioprotective properties such as angiogenesis, anti-apoptosis, anti-oxidation, metabolic reprograming, cardiac contractility, and physiological cardiac hypertrophy.

**Table 1 cells-11-04134-t001:** Effects of VEGF-B on heart.

Effects on Heart	VEGF-B Gene Transfer in Pig	VEGF-B Gene Transfer in Mouse	VEGF-B Gene Transfer in Rat	VEGF-B Overexpressed Rat	VEGF-B Knockout Rat	VEGF-B Overexpressed Mouse	VEGF-B Knockout Mouse
Angiogenesis	↑ [43,45]	ns↑ [28,44,46,47,48]	ns/↑ [39,49,50]	ns [51]	ns [39]	ns/↑ [42,51,52]	ns [17]
Arteriogenesis	N/A	N/A	ns/↑ [39,50]	↑ [39,51]	ns [39]	ns [51]	N/A
Cardiac hypertrophy	↑ [43]	↑ [44,48,52]	↑ [39,50]	↑ [39,51]	N/A	↑ [42,51]	ns/↓
Anti-apoptosis	↑ [43]	↑ [44]	↑ [50]	N/A	N/A	N/A	N/A
Lipid uptake	↑ [43]	N/A	N/A	ns [39]	ns [39]	N/A	N/A
Lipid accumulation	↑ [43]	N/A	N/A	↓ [39,53]	N/A	↑ [42]	N/A
Mitochondrial function	N/A	N/A	N/A	N/A	N/A	↓ [42]	N/A
Fatty acid β-oxidation	N/A	N/A	N/A	↓ [39,53]	N/A	N/A	N/A
Glucose metabolism	↓ [43]	N/A	N/A	↑ [39,53]	N/A	N/A	N/A
Cardiac function	N/A	ns/↑ [28,44,52]	↑ [49]	ns/↑ [51]	ns [52]	ns [42]	ns [17]
ECG alterations	Ventricular arrhythmias [54]	ns [28]	N/A	N/A	N/A	Ventricular arrhythmias [54]	Prolonged PQ interval [17]
Inflammation	N/A	mild [51]	N/A	N/A	N/A	N/A	N/A

N/A: data not available; ns: not significant.

## Data Availability

Not applicable.

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
