# Peer review of "Therapeutic Potential of VEGF-B in Coronary Heart Disease and Heart Failure: Dream or Vision?"

_cells, 2022, doi:10.3390/cells11244134_

Round 1

Reviewer 1 Report

This is an interesting paper; I just have a few remarks.

Line 102

Please consider that maybe Table 1 needs more context and explanation.

Line 105

Please consider that Figure 3 is too large and carries little information (it can be reduced to one sentence, or the figure can contain more elaborate data).

Line 283

Please consider that summary could be expanded with more finds.

Author Response

We would like to thank the reviewer for valuable feedback. We have expanded the table 1 and moved the figure 3 to summary section. 

Reviewer 2 Report

The manuscript by Mallick and Ylä-Herttuala is a comprehensive review of the role of VEGFs, and in particular VEGF-B, in their therapeutical potentials in coronary heart disease and heart failure. The laboratory of Ylä-Hertuuala is the world leading laboratory in this field and his contribution to this field is huge. I do not fully agree with the authors interpretations on some published data on mechanistic details regarding the role of VEGF-B in general, and in therapeutic angiogenesis in particular, but the present manuscript is a review of the field. The authors refer to many published papers, some of which clearly are of poor scientific quality, making the mechanistic conclusions rather weak and difficult to understand. This is not the authors fault, it is a weakness in the VEGF-B research field.

Author Response

The manuscript by Mallick and Ylä-Herttuala is a comprehensive review of the role of VEGFs, and in particular VEGF-B, in their therapeutical potentials in coronary heart disease and heart failure. The laboratory of Ylä-Hertuuala is the world leading laboratory in this field and his contribution to this field is huge. I do not fully agree with the authors interpretations on some published data on mechanistic details regarding the role of VEGF-B in general, and in therapeutic angiogenesis in particular, but the present manuscript is a review of the field. The authors refer to many published papers, some of which clearly are of poor scientific quality, making the mechanistic conclusions rather weak and difficult to understand. This is not the authors fault, it is a weakness in the VEGF-B research field.

Rebuttal: We appreciate the reviewer’s comment. As VEGF-B research field is evolving, we hope to get more new findings in future. We have also considered other interpretations and included some reservations regarding same references.

Reviewer 3 Report

In this review article, the authors discussed the potential of using VEGF-B in the treatment of coronary heart disease and heart failure. The article is up-to-date and well-written, covering the most up-to-date knowledge about VEGF-B’s role in the VEGF/VEGFR family, its splicing, binding sites, downstream signaling effects, and its key differences to VEGF-A. The authors make a strong case for the use of VEGF-B in cardiovascular diseases and cited evidence of its pro-angiogenic and other cardioprotective functions including anti-apoptosis, antioxidant, metabolic effects, and contractility. I recommend the article for publication given the minor comments below are addressed.

There is evidence that the activity of the VEGFB/VEGFR1 axis is linked to inflammation, contributing to cancer metastasis, arthritis, as well as CNS diseases such as multiple sclerosis. Much like gene therapies using VEGFA for therapeutic angiogenesis in ischemic diseases, promoting angiogenesis and inflammation could lead to vascular leakage and hyperpermeability. In Figure 1, it seems like the authors suggest that inflammation is only a problem with VEGF-A and VEGF-D. Could the authors discuss any evidence, mechanisms, or share their thoughts on VEGF-B and inflammation?

Although the article focuses on the evidence for the cardioprotective effects of VEGF-B, it would be helpful to have a section discussing its potential limitations/unwanted effects, such as susceptibility to arrhythmia, vascular inflammation, etc.

Like in coronary heart disease, using VEGF-A for therapeutic angiogenesis has also been unsuccessful in peripheral artery disease. Could the authors share any thoughts on the potential use of VEGF-B in other ischemic diseases such as peripheral artery disease?

As I understand, compared to VEGF-A and VEGF-D, VEGF-B is a much weaker inducer of angiogenesis. How does it compare to other angiogenic growth factors such as HGF, FGF?

It would be helpful if the authors could briefly introduce the expression of VEGF and VEGF receptors, especially VEGF-R1 (given the context of VEGF-B), in different relevant cell types (cardiomyocytes, vascular endothelial cells, pericytes, etc).

Author Response

  • In this review article, the authors discussed the potential of using VEGF-B in the treatment of coronary heart disease and heart failure. The article is up-to-date and well-written, covering the most up-to-date knowledge about VEGF-B’s role in the VEGF/VEGFR family, its splicing, binding sites, downstream signaling effects, and its key differences to VEGF-A. The authors make a strong case for the use of VEGF-B in cardiovascular diseases and cited evidence of its pro-angiogenic and other cardioprotective functions including anti-apoptosis, antioxidant, metabolic effects, and contractility. I recommend the article for publication given the minor comments below are addressed.

Rebuttal: We would like to thank the reviewer for valuable feedback. We appreciate your comments very much.

  • There is evidence that the activity of the VEGFB/VEGFR1 axis is linked to inflammation, contributing to cancer metastasis, arthritis, as well as CNS diseases such as multiple sclerosis. Much like gene therapies using VEGFA for therapeutic angiogenesis in ischemic diseases, promoting angiogenesis and inflammation could lead to vascular leakage and hyperpermeability. In Figure 1, it seems like the authors suggest that inflammation is only a problem with VEGF-A and VEGF-D. Could the authors discuss any evidence, mechanisms, or share their thoughts on VEGF-B and inflammation?

Rebuttal: We appreciate your comments. It’s true that VEGFB expression has been found to be linked with inflammation, contributing to cancer metastasis, arthritis as well as CNS diseases. As VEGFR1 is a known receptor for VEGF-B, that’s why it has been speculated that VEGFB/VEGFR1 axis is linked to inflammation. We have mentioned on the 2nd last sentence of Unique neovascularization in the heart section that “Actually, these results question the significance of VEGF-R1 in the VEGF-B induced angiogenic process. In addition, the current findings on the accumulation of the endothelial progenitor cells and their contribution to the neovascularization process in the heart following adenoviral gene transfer of VEGF-B186 and unsplicable VEGF-B186 (defined as “VEGF-B186R127S” due to a mutation in arginine to serine at 127 position of the amino acid sequence) as well as diminished neovascularization effect of the VEGF-B186 due to the binding to a soluble VEGF-R1 raises the possibility that there is a yet unidentified cell surface receptor for the C-terminal end of the full-length form of VEGF-B186 which could induce the neovascularization process [28,43].” We also mentioned about VEGF-B induced inflammatory effect on heart on the last sentence of Differences between VEGF-A and VEGF-B mediated effects section that “In addition, VEGF-B mediated angiogenesis and local inflammation are less prominent than those induced by VEGF-A [43,51]. ”  as well as on table 1.

  • Although the article focuses on the evidence for the cardioprotective effects of VEGF-B, it would be helpful to have a section discussing its potential limitations/unwanted effects, such as susceptibility to arrhythmia, vascular inflammation, etc.

Rebuttal: We appreciate the suggestions. Actually, we have discussed the unwanted side effects of VEGF-B at Cardioprotective role of VEGF-B section where needed, e.g.  susceptibility to arrhythmia has been discussed on Cardiac contractility subsection. In general, the mechanism of these effects has not yet been fully clarified and they are too descriptive to put under a distinct section.

  • Like in coronary heart disease, using VEGF-A for therapeutic angiogenesis has also been unsuccessful in peripheral artery disease. Could the authors share any thoughts on the potential use of VEGF-B in other ischemic diseases such as peripheral artery disease?

Rebuttal: We appreciate the interesting question. The study from our group found that VEGF-B doesn’t induce angiogenesis in either normoxic or ischemic skeletal muscle (10.1161/CIRCULATIONAHA.108.816454). So, there is very little therapeutic potency of VEGF-B in peripheral artery disease.

  • As I understand, compared to VEGF-A and VEGF-D, VEGF-B is a much weaker inducer of angiogenesis. How does it compare to other angiogenic growth factors such as HGF, FGF?

Rebuttal: This is a very good question. Unfortunately, there is no such kind of comparative study between VEGF-B and other angiogenic growth factors such as HGF, FGF. However, we think that VEGF-B is a weaker inducer of angiogenesis than HGF and FGFs. Because previous study showed that FGF-4 has quite similar angiogenic potency as VEGF-A and VEGF-D (https://doi.org/10.1096/fj.02-0377fje).

  • It would be helpful if the authors could briefly introduce the expression of VEGF and VEGF receptors, especially VEGF-R1 (given the context of VEGF-B), in different relevant cell types (cardiomyocytes, vascular endothelial cells, pericytes, etc).

Rebuttal: This is very much appreciable suggestions. We have mentioned about expression of VEGF-B and VEGF-R1 in different cells of heart. We have added “In the heart, the highest expression of VEGF-B is found in cardiomyocytes and the lowest expression is in endothelial cells [36–38]. In contrast, VEGF-R1 is predominantly expressed in cardiac endothelial cells [36–38].” at Cardioprotective role of VEGF-B section.